# AutoML Adoption in ML Software

**Koen van der Blom**                                    K.VAN.DER.BLOM@LIACS.LEIDENUNIV.NL
*Leiden University, The Netherlands*

**Alex Serban**                                                    A.SERBAN@CS.RU.NL
*Radboud University, The Netherlands*

**Holger Hoos**                                                        HH@LIACS.NL
*Leiden University, The Netherlands – University of British Columbia, Canada*

**Joost Visser**                                    J.M.W.VISSER@LIACS.LEIDENUNIV.NL
*Leiden University, The Netherlands*

## Abstract

Machine learning (ML) has become essential to a vast range of applications, while ML experts are in short supply. To alleviate this problem, AutoML aims to make ML easier and more efficient to use. Even so, it is not clear to which extent AutoML techniques are actually adopted in an engineering context, nor what facilitates or inhibits adoption. To study this, we define AutoML engineering practices, measure their adoption through surveys, and distil first insights into factors influencing adoption from two initial interviews. Depending on the practice, results show around 20 to 30% of the respondents have not adopted it at all and many more only partially, leaving substantial room for increases in adoption. The interviews indicate adoption may in part be inhibited by usability issues with AutoML frameworks and the increased computational resources needed for adoption.

## 1. Introduction

Recent advances in machine learning (ML) have resulted in increasingly widespread use of ML methods in a wide range of applications. AutoML promises to decrease development time to enable adoption of ML on a larger scale and also to empower developers less experienced with ML. By automating key stages in creating ML tools and pipelines, AutoML trades human time and experience for compute time. However, it is not yet clear to which extent developers adopt AutoML techniques, and what factors influence their decisions.

To measure adoption we formulate practices based on a selection of AutoML techniques. For the three AutoML practices we consider, the results from two surveys with a combined total of over 360 development teams are presented. Moreover, we present preliminary results from interviewing ML practitioners, which add depth to our survey results.

We investigate the adoption of AutoML practices in the broader context of software engineering for ML (SE for ML), where AutoML practices automate different stages of the software development life cycle for ML. Automation is known to bring multiple benefits in traditional software development. However, its impact on the development of ML systems is still unknown. Along with other results, we previously collected data about automatic hyperparameter optimisation from over 300 developers (Serban et al., 2020). With an extended survey, additionally covering trustworthy ML, and also including additional practices on automatic feature generation and model configuration, we gathered more than

60 responses so far (Serban et al., 2021). Here, we analyse the AutoML data from these quantitative studies to obtain a fine-grained view of AutoML practices and initiate qualitative interviews with practitioners to situate the survey results. Overall, we observe the adoption of AutoML practices is relatively low, and the preliminary interviews suggest this may in part be motivated by usability issues with AutoML frameworks and the increase in computational resources required by AutoML approaches.

To position this study, Section 2 covers related work. Next, Section 3 describes the AutoML practices considered in our work, and discusses the survey design and interview setup. Section 4 analyses the AutoML results from Serban et al. (2020) and from the extended survey from Serban et al. (2021), and relates them to observations from the interviews. Lastly, in Section 5, we briefly summarise our main findings and discuss directions for future work.

## 2. Related Work

Best practices have been defined for research and for benchmarking of AutoML techniques and frameworks. For instance, Eggensperger et al. (2019) define best practices for the use of automatic algorithm configuration. Similar work exists for other techniques, such as neural architecture search (Lindauer and Hutter, 2020). Best practices to compare AutoML frameworks have also been proposed, e.g. by Gijsbers et al. (2019) combined with a set of benchmark problems. However, none of this prior work considers the practical application of AutoML, or AutoML practices used by developers of applications with ML components.

In the field of human computer interaction, there is recent work with a focus on the adoption and accessibility of AutoML. Wang et al. (2019) postulate that users may not adopt AutoML, because they distrust AutoML systems – for instance, because they do not know whether the system was run sufficiently long. To tackle this problem, they introduce a system to visualise and analyse results to help the user understand what the system does, and to guide it. Lee et al. (2019) suggest that AutoML adoption may be low due to usability issues rather than a lack of awareness, but do not indicate any evidence for this. They propose a mixed-initiative ML framework with the aim of improving the usability which should, among other things, result in a more transparent framework that users are more likely to trust. Drozdal et al. (2020) investigate which features are important to gain the trust of AutoML users, and found that visualising the process and providing performance metrics are the most important. While these works aim to improve usability or trustworthiness in order to facilitate increased adoption, they do not actually establish that adoption is low, or that these are the primary factors limiting adoption.

In addition to the usability of AutoML frameworks themselves, there is also work on making AutoML frameworks more usable by simplifying the interaction with them, or to extend the automation to earlier or later steps than handled by the current frameworks. For instance, Cambronero et al. (2020) introduce a tool to help define the search space for AutoML frameworks. With another tool, Narkar et al. (2021) aim to help users with the analysis of results and to let them choose a final model based on more than just a performance metric optimised by the AutoML framework.

As far as actual AutoML adoption goes, Lee et al. (2019) observe, based on preliminary results, that fewer than 2% of the workflows available on OpenML (Vanschoren et al., 2013) adopt AutoML. However, since OpenML is primarily a research platform, this may not

be a very good indicator for adoption in a wider context; in addition, it gives no detailed view of adoption of the various AutoML stages. Xin et al. (2021) study for which parts of the ML process AutoML adopters apply the automated methods and which they prefer to do manually. Their findings also cover the benefits and shortcomings the users see in the current AutoML frameworks. For the steps where AutoML is used most (hyperparameter tuning, model- and feature selection), however, they go into the least detail, merely stating that these are the most used; insights into what drives or holds back adoption of these steps are not reported. In a similar study Crisan and Fiore-Gartland (2021) look more closely at the use and adoption of visualisation techniques for AutoML, and how desirable users actually find them. Both Xin et al. (2021) and Crisan and Fiore-Gartland (2021) gather qualitative data from interviews and suggest that human experts (currently) demand a place in the loop because AutoML does not sufficiently address all their needs.

## 3. AutoML Practices

**Practice mining.**  To extract the practices, we followed a similar process to Serban et al. (2020), resulting in three AutoML practices, which overlap with the pre-processing and modelling tasks from Xin et al. (2021) that are well established in academia. These practices were derived from grey literature as follows. Dean (2019), Tunguz (2020) and ZelrosAI (2019) all consider various AutoML stages, which can be categorised as revolving around (a) feature engineering and selection, (b) hyperparameter optimisation (HPO) and model or algorithm selection, and (c) the configuration of algorithms or model structures and architectures. This categorisation resulted in the following practices: [1]

1. Automate feature generation or selection.

2. Automate model selection and hyperparameter optimisation.

3. Automate configuration of algorithms or model structure (e.g., NAS).

**Practice adoption.**  To measure the adoption of these AutoML practices, we ran two surveys in the general context of software engineering for ML, i.e., the surveys contained both questions about software engineering for ML and AutoML. The questionnaire was designed following the recommendations from Kitchenham and Pfleeger (2002) as a cross-sectional observational study, asking participants at the moment of filling the questionnaire if they adopted the practices. Several preliminary questions provided the basis for assigning participants to pre-defined groups. Participants were instructed to answer from the perspective of their team, rather than based on their individual practices.

The questions (see Appendix A) used standard answers on a Likert scale with four possible answers reflecting degrees of adoption rather than levels of agreement. This allowed the practices to be expressed impartially and avoided the null-point bias strategy. The four possible answers were "Not at all", "Partially", "Mostly" and "Completely".

The first survey covered only the second AutoML practice mentioned above, while the second survey was specifically designed to cover all practices. Details on how the survey was distributed and a demographic analysis to identify possible biases can be found in Serban et al. (2020). Other SE for ML practices are also available there and in Serban et al. (2021).

---

1. More detailed descriptions are available in our practice catalogue: `https://se-ml.github.io/practices/`

**Interviews.** To complement our survey results, but also to validate that the practices we identified are indeed relevant for practitioners and developers of AutoML frameworks, we conducted interviews. The interview protocol was designed using the guidelines from Hove and Anda (2005) and consisted of 15 questions designed to support a natural conversation.

The interviews were structured as follows. First, we described the goals and background of our research. Next, we asked participants to share information about their background and demographics. We then asked participants about the way they use AutoML and about the challenges faced by them in adopting AutoML, the perceived benefits and the risks associated with the use of AutoML tools and techniques. Finally, we asked participants open-ended questions designed to elicit additional thoughts and feedback from them.

We recruited participants using purposeful sampling (Palinkas et al., 2015), by contacting practitioners which we knew were using or interested in using AutoML techniques. We present results from two interviews. Participant **P1** works on applications with ML components at a European technology company, while participant **P2** works on AutoML at a European university (not our own). Both have five or more years of experience with ML.

## 4. Results

Since the two surveys were open during separate time frames, and adoption may have changed, we also analyse their results separately. Furthermore, we discuss the interviews in relation to the survey results whenever there is a clear connection, followed by further observations from the interviews. Additional survey results are available in the appendices.

**Survey 1.** The initial survey resulted in a total of 307 valid responses[2] to the question on model selection and hyperparameter optimisation. Out of these respondents, 8% reported to have completely adopted these AutoML techniques, 26% mostly, 35% partially, and 31% not at all (see Table 1 in Appendix B). Evidently, although these techniques appear to be widely known, there is still significant room to increase adoption. Adoption levels between geographic regions (Figure 3a, Appendix C) are quite similar, with only South America showing higher rates of adoption, which are likely the result of the comparatively small size of this particular sample (6% of the respondents).

Figure 1a breaks down adoption by organisation type. Adoption is highest for research laboratories, followed closely by technology companies. Non-tech companies as well as governmental organisations show substantially lower adoption. Part of the explanation for this difference may be the proximity of research and tech organisations to the latest technological developments. However, it seems unlikely that this is the only factor. Non-tech companies and governmental organisations may work with sensitive data more frequently, which could make them more cautious about adopting AutoML techniques. Usability issues are another factor that could play a role, as also postulated by Lee et al. (2019), and these may be easier to overcome for people with a strong technical background.

There appears to be a positive correlation between experience and adoption (Figure 1b). Specifically, a substantial change in adoption appears after two years of experience. Having more than five years of experience does not seem to result in an additional increase in adop-

---

2. Note that numbers in analysis may be slightly lower due to respondents that did not answer all questions. Data, questions, and code from this survey are available at: `https://doi.org/10.5281/zenodo.3946453`

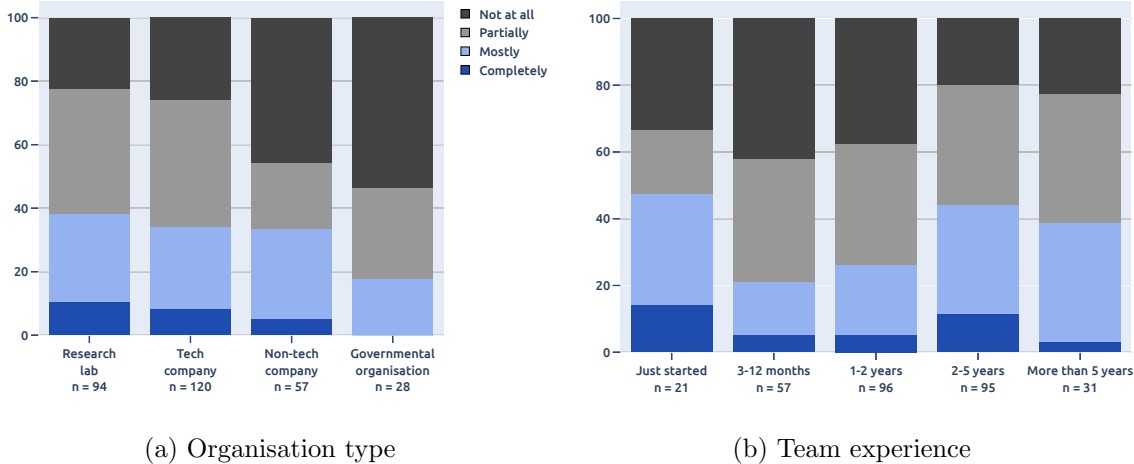

(a) Organisation type          (b) Team experience

Figure 1: Model selection and hyperparameter optimisation.

tion. These results are somewhat at odds with the notion that AutoML may democratise ML by making it more accessible to users with less ML expertise. A comment by interview participant P2 (academic) could provide part of the explanation why more experienced teams may have higher adoption. They indicated that it is difficult to judge how long AutoML frameworks should be allowed to run to get good results while avoiding overfitting, something that might be learned with experience. This was echoed by P1 (tech), who indicated being more concerned about overfitting as a result of using AutoML techniques compared to using a manual approach based on domain knowledge. The respondents who just started using ML appear to be outside the perceived trend, but since they represent a relatively small group of the answers (7%), this result may be statistically less reliable.

**Survey 2.** So far, the second survey resulted in 67 valid responses[3] to the AutoML questions. Although the number of responses is much lower than in our first survey, the breakdown into three questions concerning AutoML already allows us to gain additional insights. In Figure 2, the adoption per practice is shown. Automated configuration of algorithms and the structure of models is clearly less adopted than the other two practices. For feature generation and selection, it is interesting to note that more than 20% of the cases do this implicitly – for instance, by using deep learning. The number of respondents that adopted model selection and hyperparameter optimisation at least partially is higher than in the first survey, leaving only around 20% that do not adopt this practice at all. Due to the lower number of responses from our second survey, it is still difficult to draw meaningful conclusions regarding adoption differences between organisation types and other demographics.

**Interviews.** In addition to the factors influencing AutoML adoption mentioned so far, the interviewees also raised some other interesting points. Specific to automated feature selection, P1 (tech) saw few issues, raising that they can look back at what the selected features mean and whether the selection makes sense. P1 also indicated that the initial required investment to adopt new AutoML techniques is high, and at odds with short

---

3. Data form this survey will be made available at a later stage. See Appendix A for the AutoML questions.

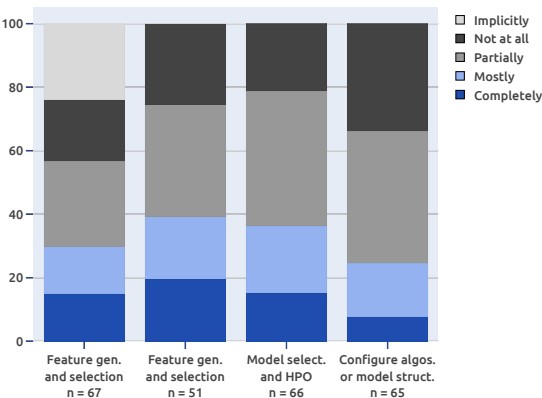

Figure 2: Adoption per practice. The first bar shows adoption of the feature practice including *'Implicit, e.g. deep learning'* answers, while in the second bar they are excluded.

production times to ship to customers. However, P1 did see potential benefits in the long run, where AutoML might provide savings in the time spent by human experts. Since with deep learning, explainability is already an issue, P1 was not too concerned about possible explainability issues stemming from AutoML use in this context.

P2 (academic) mentioned that in cases where AutoML frameworks don't work, it is generally difficult to find out why it is not working. In which component is the issue? Is it a problem with the dataset or a bug? For P1, neural architecture search on the full search space would be too expensive, making the manual design with expert knowledge more practical. P1 further indicated that the use of ML (without AutoML) often already led to improvements for their customers; in combination with limited production time, in their view, that left relatively little opportunity and motivation for using AutoML techniques to possibly achieve improved performance.

## 5. Conclusion

Aiming to measure AutoML adoption by practitioners working on applications with ML components, we identified three engineering practices for AutoML, and measured their adoption through surveys. In addition, a preliminary set of interviews was held to gain insight into the factors driving and inhibiting adoption. Our findings indicate that there is substantial room to increase adoption across the board. Based on the preliminary interviews, the relatively low adoption might result from usability issues with AutoML frameworks and the substantial computational resources required for AutoML.

In the near future, we plan to conduct additional interviews. So far, our two interviews only provided first insights into possible reasons for limited adoption of AutoML practices. It would be of particular interest to interview people that have not yet adopted AutoML, since both the interviews we present here and those by Xin et al. (2021) targeted users with at least partial adoption. We expect that this will help us in testing and refining our preliminary ideas regarding the reasons why some ML practitioners refrain from adopting AutoML tools and techniques.

## Acknowledgements

This research was partially supported by TAILOR, a project funded by EU Horizon 2020 research and innovation programme under GA No 952215.

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

# Appendix A. Survey Questions about AutoML

## A.1 Survey 1

1. We can perform model selection and hyper-parameter optimization in an automated way.

## A.2 Survey 2

1. We use automated methods to generate or select features from input data.

2. We perform model selection and hyper-parameter optimisation in an automated way.

3. We use automated methods to configure our algorithms or the structure of our models.

# Appendix B. Survey 1 – Overall Model Selection or Hyperparameter Optimisation

| Answer | Number | Percentage |
|---|---|---|
| Not at all | 96 | 31% |
| Partially | 106 | 35% |
| Mostly | 80 | 26% |
| Completely | 25 | 8% |

Table 1: Overall adoption

# Appendix C. Survey 1 – Model Selection or Hyperparameter Optimisation by Demographic

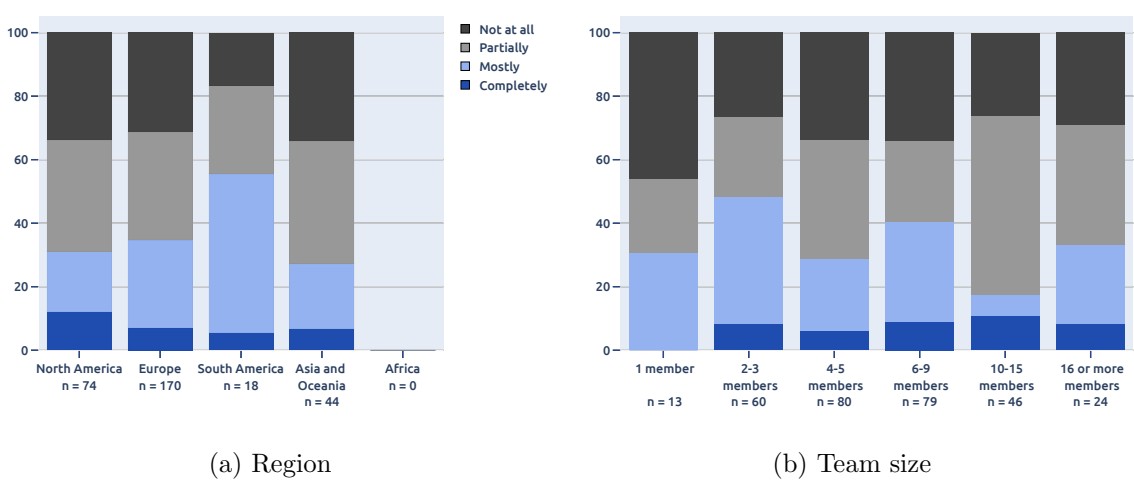

(a) Region

(b) Team size

Figure 3: Model selection or hyperparameter optimisation.

## Appendix D. Survey 2 – Mean Adoption by Demographic

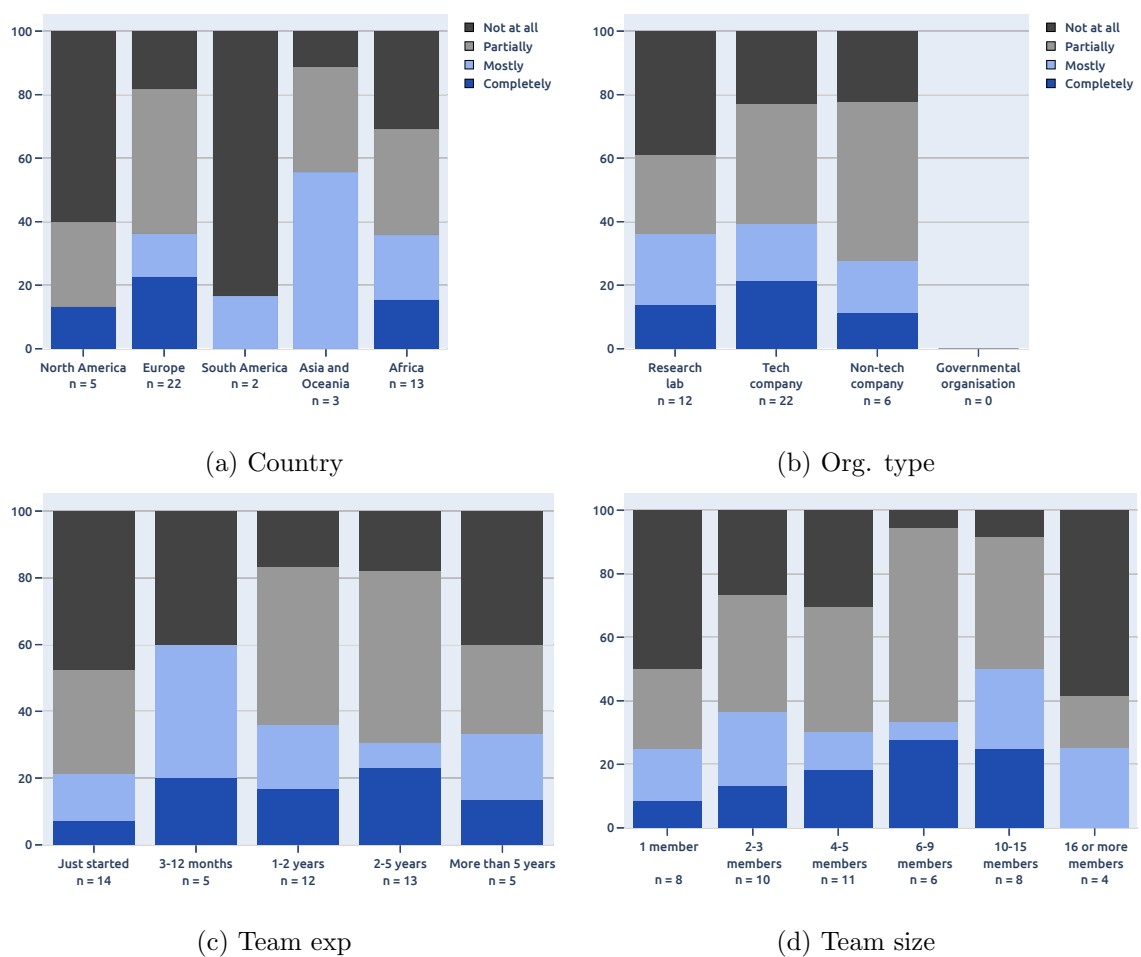

(a) Country

(b) Org. type

(c) Team exp

(d) Team size

Figure 4: Mean over all three AutoML practices, excluding responses that answered 'Implicit, e.g. DL' to feature generation/selection.

## Appendix E. Survey 2 – Feature Generation/Selection by Demographic

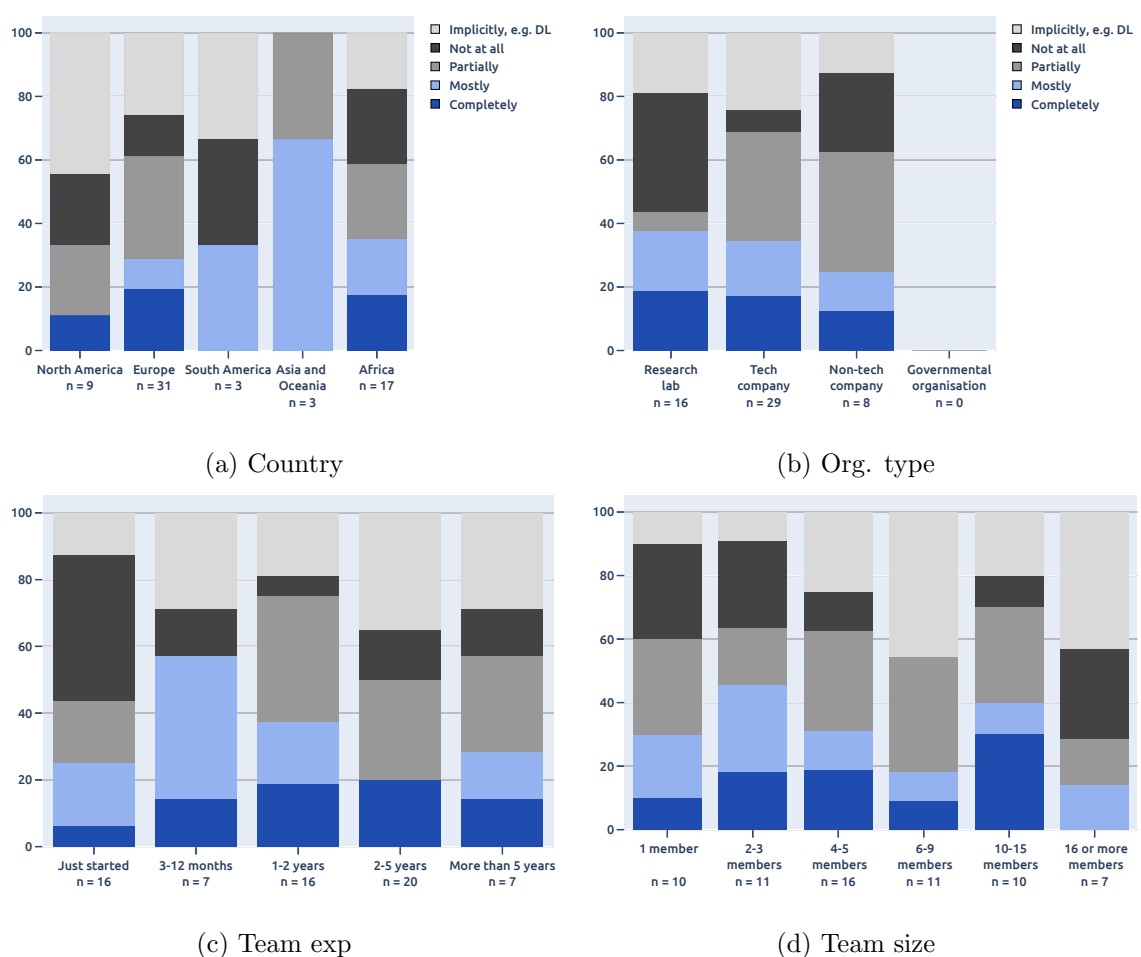

(a) Country

(b) Org. type

(c) Team exp

(d) Team size

Figure 5: Feature generation/selection.

## Appendix F. Survey 2 – Model Selection or Hyperparameter Optimisation by Demographic

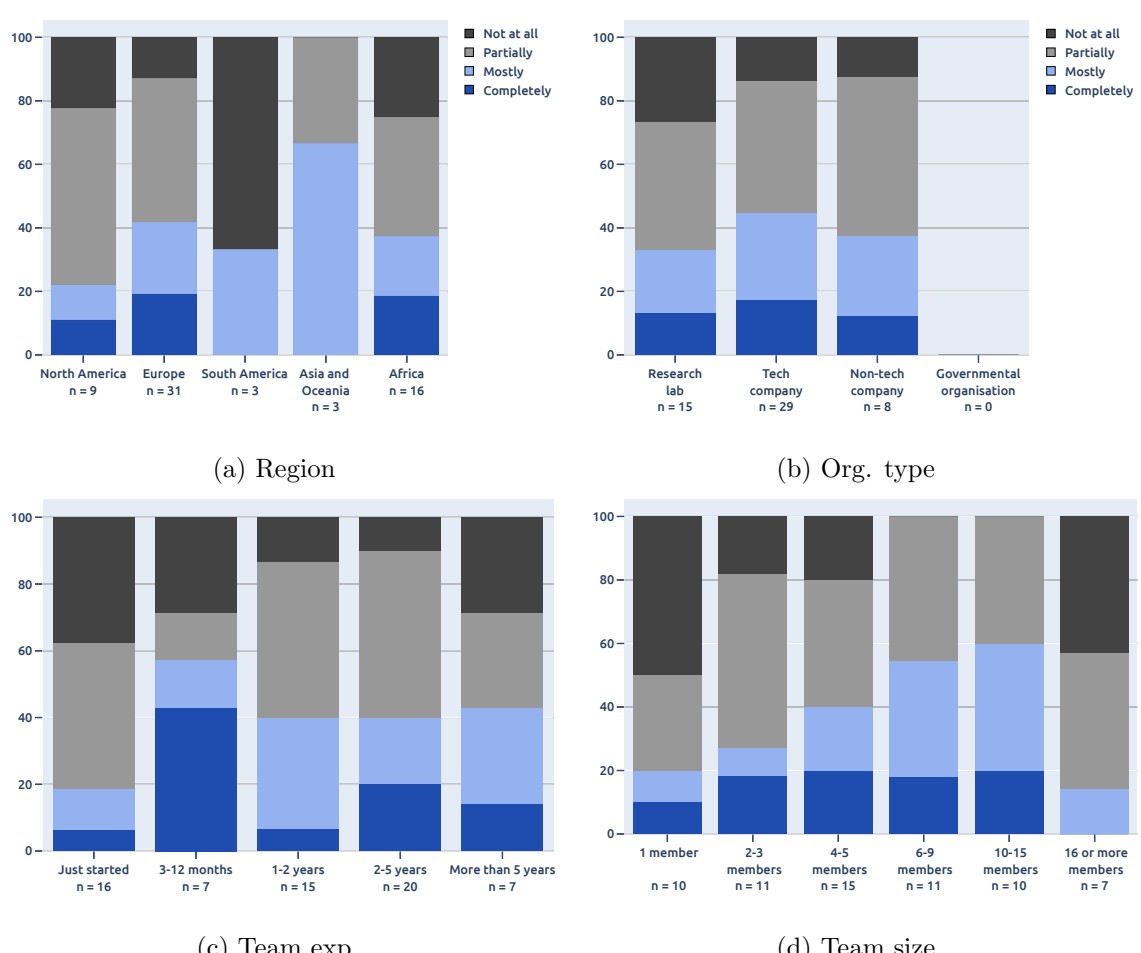

(a) Region

(b) Org. type

(c) Team exp

(d) Team size

Figure 6: Model selection or hyperparameter optimisation.

## Appendix G. Survey 2 – Algorithm/Model Configuration by Demographic

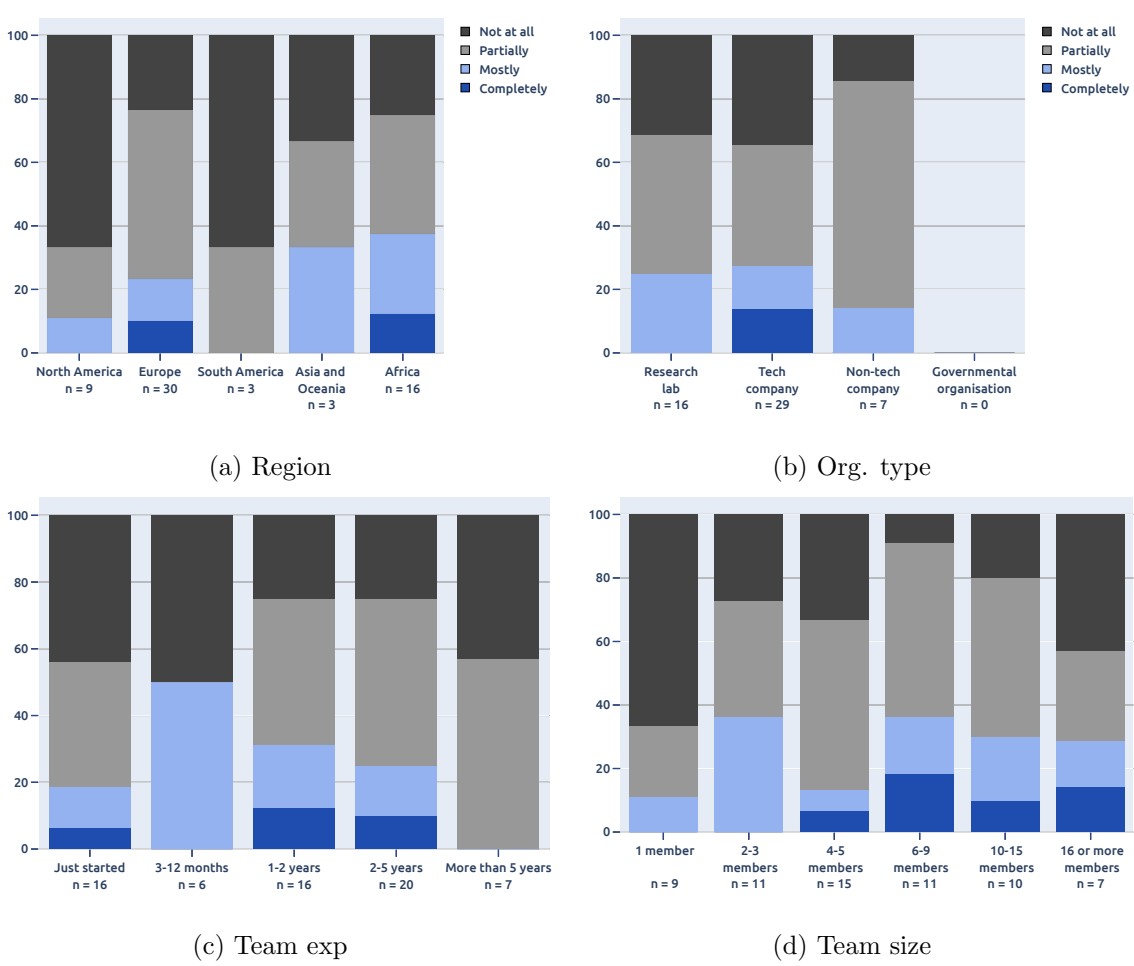

(a) Region

(b) Org. type

(c) Team exp

(d) Team size

Figure 7: Algorithm/model configuration.

