# OpenReview forum: "AutoML Adoption in ML Software"
_ICML.cc/2021/Workshop/AutoML — AutoML@ICML2021 Poster_

### Official Review · Reviewer_Gx77 · 2021-06-10
**I think it is an ok paper with an expected message that is now scientifically a bit more sustained.**

**Rating:** 6
**Confidence:** 4

**Review:**

The paper addresses the question of adoption of AutoML in ML-oriented software development. I consider this a very important question, because it establishes an indicator of the extent to which AutoML achieves its goal. AutoML was meant to support humans. If humans don't use it, AutoML is obsolete.

Now there are some remarks I have on the paper

pros:
- addressed question is very important
- the methodology seems generally principled
- it is a good idea to complement the survey with interviews
- in a  community that is  overly driven by model selection, the paper puts over the table the important topics of acceptance and interaction (including usability) already introduced by some few other authors earlier.

cons:
- I would insist on a detailed description of how the population was reached. it is very nearby to suspect that the sample of participants is in a close neighborhood of the authors, which can introduce a substantial bias. The population is, in my understanding, the set of all software developers (world-wide) that are involved in the development of ML software. How can you argue that your sample is representative for this population?
- assertions about correlations should not be justified with figures but with numbers.
- the paper is somehow not very readable. I cannot trace it back to one particular thing, but I had to read the introduction three times to understand it. Maybe some aspects are very abstract in the beginning. For example, you talk about "three practices you consider", but without knowing what is meant by practices and without naming the three you look at, it is hard to give semantics to those sentences.

another remark that is not directly a con of the paper but of the approach in general is that, in my view, software developers are probably not even the population that one would expect to use AutoML primarily. Instead, I would see data science consultants as the primary group.

---

### Official Review · Reviewer_8AaX · 2021-06-16
**Interesting survey for AutoML researchers**

**Rating:** 7
**Confidence:** 3

**Review:**

This paper presents the results of 2 surveys on AutoML engineering practices. The results aligns with other surveys showing that most practitioners do not use AutoML tools, but it does not provide insights on why it is the case.

Pros:
- Important topic. AutoML research is undervalued if there are low adoption rates in academia and industry of the tool produced.
- Good coverage of related works.
- Nuanced interpretation of the results.
- Clear, well written.

Cons:
- It is not clear what the exact questions were in the survey. An example of the survey should be provided in the appendix.
- Few insights on why AutoML tools have low adaption rates.
- Many of the results are inconclusive. Also there are no confidence intervals reported and therefore the statistical significance is evaluated rather imprecisely (but nuanced appropriately).

I do not know what is the position of the workshop organizers for surveys, if they believe it is a right fit for a workshop paper. To be clear this is not a survey of literature, it is a survey of experience and practices of a group of people. I believe this kind of survey is very useful to self-reflect on our field and ensure our research has an actual impact in real world problems, so I would suggest acceptance.

---

### Decision · Program_Chairs · 2021-06-21

Accept (Poster)